# Advances in Nucleic Acid Universal Influenza Vaccines

**DOI:** 10.3390/vaccines12060664

**Published:** 2024-06-17

**Authors:** Liang Xu, Weigang Ren, Qin Wang, Junwei Li

**Affiliations:** 1Department of Infectious Disease, The Second Hospital of Nanjing, Affiliated to Nanjing University of Chinese Medicine, Nanjing 210003, China; luomu77956@163.com (L.X.); rwg20161564@163.com (W.R.); wangqin6621628@163.com (Q.W.); 2Medical Innovation Center for Infectious Disease of Jiangsu Province, Nanjing 210003, China

**Keywords:** DNA vaccine, mRNA vaccine, influenza virus, universal influenza vaccine

## Abstract

Currently, vaccination with influenza vaccines is still an effective strategy to prevent infection by seasonal influenza virus in spite of some drawbacks with them. However, due to the rapid evolution of influenza viruses, including seasonal influenza viruses and emerging zoonotic influenza viruses, there is an urgent need to develop broad-spectrum influenza vaccines to cope with the evolution of influenza viruses. Nucleic acid vaccines might meet the requirements well. Nucleic acid vaccines are classified into DNA vaccines and RNA vaccines. Both types induced potent cellular and humoral immune responses, showing great promise for the development of universal influenza vaccines. In this review, the current status of an influenza universal nucleic acid vaccine was summarized.

## 1. Introduction

Seasonal and pandemic influenza viruses, as well as emerging zoonotic influenza viruses, cause intense concern for public health. According to the report by the World Health Organization (WHO), annual seasonal influenza virus epidemics lead to approximately 3–5 million infections and 290,000–650,000 deaths worldwide [1]. Pandemics of influenza viruses caused even more serious damage and death in the historically recorded pandemics, including the 1918 (H1N1) Spanish Flu, 1957 (H2N2) Flu Pandemic in Asia, 1968 (H3N2) Hong Kong Pandemic, 1977 (H1N1) Russia Flu Pandemic and 2009 (H1N1) Flu Pandemic around the world [2,3]. Avian-origin influenza viruses spillover could also result in potential epidemics, even pandemics, and lead to public panic, such as the H5N1 influenza virus with around 50% mortality and H7N9 with 40% mortality, despite lacking evidence of human-to-human transmission [4,5]. 

Currently, vaccination with seasonal influenza vaccines is an effective strategy to prevent seasonal pandemic or endemic influenza viruses. Influenza virus vaccines licensed include inactivated, recombinant and live-attenuated influenza vaccines (LAIVs), which contain antigens from H1N1, H3N2 and two influenza B types, and mainly induce humoral immune responses against the viral surface glycoproteins, hemagglutinin (HA) and neuraminidase (NA) [6]. Due to the high variation of both surface proteins of influenza viruses, the protection efficacy conferred by vaccination with seasonal influenza vaccines is limited [7]. Statistically, the effectiveness of current influenza vaccines is under 60% [8]. Inactivated and recombinant influenza vaccines fail to trigger a vigorous immune response in those 65 and older and immuno-compromised people, who are vulnerable when facing an infection by seasonal or emerging influenza viruses [9]. 

LAIVs were reported to provide strong, long-lasting cell-mediated and potent humoral immunity; nevertheless, the biological safety of LAIV is a major concern. The fear of virulence reversal and viral shedding by LAIV is a key hurdle preventing the broad distribution of LAIV [10]. Egg-based influenza virus vaccines cause antigen changes because of the possibility of generating chicken embryo adaptive mutations or protein posttranslational modifications, which enhance the replication of vaccine virus in eggs, thus leading to an antigenicity shift. In addition, there is a time-gap between the influenza viruses recommended by WHO influenza surveillance system and their production, leading to untimely supplies of influenza vaccine following the reports of seasonal influenza surveillance.

In recent years, many approaches have been employed to eliminate the defects of existing influenza vaccines to overcome the antigen drift and shift caused by the evolution of influenza viruses, including the development of vaccines covering as many strains as possible and the adaption of highly conserved epitopes adopted from HA2, M2e, or NP of different influenza viruses [11]. In spite of the high specificity of humoral immune responses that bind and neutralize influenza viruses, T cell immune responses instigated by T cell-specific epitopes or adjuvants show cross-reactivity in a broad manner to control infection by influenza viruses [12]. Compared to seasonal influenza vaccines that are composed of multivalent HA protein antigens or live-attenuated influenza viruses, nucleic acid influenza vaccines encoding influenza virus antigens inducing potent humoral and cellular immune responses could enhance the protection against a broad array of influenza viruses [13].

In order to better elucidate the mechanism of nucleic acid vaccine reactions in the body, the process of the immune response induced by nucleic acid influenza vaccines is illuminated in Figure 1. As shown, after encapsulated DNA or mRNA enters target cells, the particle is dissociated in endosomes with an acidic environment, releasing the antigen-coding DNA or mRNA into the cytoplasm. mRNAs as templates are translated by the host protein synthesis machinery directly in the host cytoplasm. Otherwise, different from mRNA, the DNA is first transported to the nucleus, then transcribed into mRNA, which is necessary for the antigen expression, then used as a template to be translated into proteins. The translated antigenic protein is processed by lyases to generate antigenic peptides that bind to the major histocompatibility complex (MHC I or II). The bound complex is presented on the cell surface, recognized by receptors on the ancestral B cells or T cells, and elicits an immune response. 

The humoral immune response is mainly mediated by B lymphocytes, which recognize antigens presented via the MHC II pathway and are activated through the synergistic action of T helper (Th) cells. Activated T cells secrete IL-4 and IL-21, which promote the entry of B cells into the germinal center and their differentiation into plasma cells or memory B cells. Plasma cells are responsible for producing and secreting specific antibodies to neutralize influenza viruses. Memory B cells, on the other hand, provide a rapid and potent antibody response to the secondary immune response by stimulation of the same antigenic peptides. 

The cellular immune response involves two subtypes of T cells. Helper T cells enhance the overall immune response by recognizing antigens and producing cytokines (e.g., IL-2 and IFN-γ) via the MHC II pathway, while CD8+ cytotoxic T cells recognize and destroy virus-infected cells expressing pathogen-specific antigens via the MHC I pathway. In this process, cytokines have indispensable functions in orchestrating cellular interactions and shaping the nature and magnitude of immune responses. 

Researchers have demonstrated that DNA and mRNA themselves present immunogenicity or adjuvanticity, which instigates innate immunity and induces cytokines, enhancing the cellular and humoral immune responses and vice versa. The diagram also illustrates the inherent memory function of the immune system, which is reflected in the generation of memory T cells that rapidly respond to previous infectious events and protect against future infections [14,15].

As for a preventive approach to infectious diseases, the development of prophylactic strategies against hypervariable infectious pathogens should be focused on multiple targets or antigens. In the case of influenza viruses, there has been much progress in recent years in novel and rational universal nucleic acid influenza vaccine designs, but so far, no effective one has been approved for use for this purpose. However, the advances in this field have brought the light of hope. In a recent published paper of a phase 1/2 randomized clinical trial of mRNA-based seasonal influenza vaccine, the results showed that an mRNA vaccine (mRNA-1010) elicited a potent humoral immune response [16]. Here, universal nucleic acid influenza vaccines in development are summarized as a new vaccine platform that shows several advantages against the conventional vaccine platform based on protein antigens.

## 2. A Brief Description of Influenza Viruses

### 2.1. Influenza Epidemiology

There are two different influenza seasons in the world. In the Northern Hemisphere, it is between November and March. In the Southern Hemisphere, it is between June and September. Influenza can be classified into pandemics, epidemics, localized outbreaks, and sporadic cases infected by avian-origin influenza viruses, according to the degree of epidemic. Seasonal influenza is an acute respiratory infection caused by influenza A or B viruses that is endemic around the world with symptoms ranging from mild to severe and potentially fatal. All age groups could be affected. Symptoms of influenza include acute onset of fever, cough, sore throat, body aches and fatigue. Hospitalizations and deaths occur mainly in high-risk groups such as young children, pregnant women, the elderly, and immune-compromised persons [17]. In most adults, the symptoms are mild and they recover within 1–2 weeks. There are also reports on the spillover of avian influenza viruses from avians to human, such as the constantly reported infections by H5N1 influenza viruses with the potential for human-to-human transmission. The H7N9 epidemic in 2013 caused over 1500 infections, and H3N8 infections have been reported in humans [1,4]. These cross-species transmissions of avian influenza viruses led to public panic when they were reported in an exaggerated manner.

### 2.2. Influenza Virus Typology and Structural Characteristics

There are four types of influenza viruses according to their NP gene, classified A, B, C, and D [18]. Furthermore, influenza A viruses are subtyped according to the hemagglutinin (HA) and neuraminidase (NA) on the surface of the influenza virus. There are currently 18 different H subtypes (H1-H18) and 11 different N subtypes (N1-N11) of influenza A viruses [19]. The subtypes of influenza A virus that often circulate in the population include A/H1N1 and A/H3N2. Influenza B viruses that circulate in humans are not classified into subtypes but are further divided into two lineages, namely B/Yamagata and B/Victoria.

The influenza A and B viruses have a segmented genome coding 10–12 proteins. The polymerase is responsible for replication and transcription of these segments. The PB1 subunit contains the catalytic activity for RNA synthesis, while PB2 is involved in cap-binding and host adaptation. PA is responsible for endonuclease activity and cleavage of host mRNA [20]. In the influenza virion particles, the polymerase forms a complex with the viral RNA (vRNA) and nucleoprotein (NP) to assemble the ribonucleoprotein (RNP) complex, which is essential for viral replication and protects the viral RNA from degradation by RNases [21]. There are four structural proteins on the viral envelope, HA, NA, M1 and M2, which were identified as viral antigens to be used in vaccine development [22,23,24].

Similar to influenza A, the genome of influenza B, comprising eight negative-sense, single-stranded viral RNA (vRNA) segments, encodes four envelope proteins, HA, NA, NB, and BM2, and ribonucleoproteins PB1, PB2, PA, and NP [25]. Analysis of the evolutionary dynamics showed Yamagata lineage viruses had alternating dominance between antigenic groups, while Victoria lineage viruses showed antigenic drift of a single lineage [26].

The HA trimer is synthesized as a single polypeptide chain (HA0). HA0 is then cleaved by host cell proteases to become mature HA, which consists of HA1 and HA2 subunits. HA1 promotes viral entry by engaging the receptor and HA2 mediates virus-host membrane fusion [27]. NA is an essential glycoprotein on the surface of the influenza virus and it is responsible for the release of progeny virions from the host cell to infect new cells [28]. M2 protein is an ion channel that plays a crucial role in viral uncoating and replication. Upon entering the acidic environment of the endosome, M2 facilitates proton influx into the virion, leading to the disassociation of vRNPs from the viral matrix protein (M1) and their subsequent release into the host cellular cytoplasm [29]. The M1 protein is the most abundant protein in the influenza virion and it plays a critical role in viral assembly and budding by providing structural integrity and shaping the viral particles [30]. 

NP encapsulates the vRNA, protecting it from degradation and recognition by the host’s innate immune system, and it has been identified as a conserved internal antigen, primarily targeting cytotoxic T lymphocyte (CTL) recognition during influenza virus infection and it has been shown to mediate cross-protection against heterotypic influenza viruses [31,32]. M2e is the extracellular N-terminal domain of M2 that is highly conserved in human influenza A virus strains, mediating protective immunity and relying primarily on Fcγ-mediated effector mechanisms, such as antibody-dependent cell cytotoxicity (ADCC) or the anti-immunogenicity of M2e. Attempts have been made to couple the M2e domain to vectors and novel adjuvants and multiple immunizations with high doses have been used to enhance its antigenicity [33].

HA2 forms most of the highly conserved stem-like structure, which anchors the globular domain to the viral membrane and contains the viral fusion peptide, inducing a cross-protective immune response [34]. The long α helix (LAH) domain (HA2 76-130aa) is the most conserved region in the HA stem region of the influenza virus, which has also been explored as a target for broad-spectrum influenza vaccines. Antibodies against stems typically have broad cross-reactivity and often have a neutralizing effect, and they can also help eliminate infected cells through Fcγ-mediated effector function (ADCC) [35,36].

## 3. Nucleic Acid Vaccines

Nucleic acid vaccines use genetic material from a disease-causing virus or pathogenic bacteria to stimulate an immune response against these pathogens. Nucleic acid vaccines have two forms, DNA vaccines and RNA vaccines [15]. DNA vaccines are designed on the base of plasmids that encode the targeting antigens, such as HA, NA and M2 of influenza viruses, which stimulate humoral immune responses and sometimes induce immununostimulatory molecules. The common immunostimulatory molecules used as adjuvants include IL-2, GM-CSF, CpG, etc. [37]. DNA vaccines not only stimulate humoral immune responses but also cellular immune responses. There are several routes employed as administration methods of DNA vaccines, including intramuscular (IM), intradermal (ID), mucosal, and electro-transportation [38]. In addition, DNA vaccines have been approved to prevent infectious disease in animals, such as horse West Nile virus disease [39].

In the 1990s, in vitro transcribed mRNA was shown to be transported into animal cells and be translated in mice [40]. Since then, mRNA technology was introduced into vaccine research and development. After internalization by endocytosis, a DNA vaccine is transferred to the nucleus for transcription, and the transcribed mRNA is exported into the cytoplasm for translation. However, RNA vaccines are translated directly after being released from endosomes [41].

### 3.1. Advantages and Disadvantages of Nucleic Acid Vaccines

Researchers suggested that nucleic acid vaccines showed the features of simplicity, high safety, effectiveness and low cost. DNA and RNA vaccines are non-living entities without the risk of transformation into a pathogen, which enhances their safety during vaccine manufacture. Furthermore, both of them induce potent T cell and humoral immunity and can be manufactured on a large scale. In recent studies, the results showed that mRNA vaccines could be delivered intranasally and instigated potent mucosal immune responses including T cell and humoral immune responses, which are crucial to inhibit the replication of respiratory pathogens [42,43,44]. However, there are still some disadvantages within nucleic acid vaccines that should be resolved. DNA vaccines showed relatively low immunogenicity, especially in large animals. As mentioned above, during the process of antigen expression, DNA enters the cellular nucleus, which may incur risks with the potential to integrate into the host genome, leading to insertional mutagenesis [41,45]. As well, the formation of anti-DNA antibodies, induction of autoimmunity and immunological tolerance are the concerns in the design of DNA vaccines.

### 3.2. Universal Influenza DNA Vaccines

Compared to traditional influenza vaccines, influenza vaccines based on nucleic acids have significant advantages, bringing together the nucleic acids encoding the antigens of each influenza virus strain, creating a multivalent universal influenza vaccine or conserved antigen vaccine, which would solve the problem of the differences between the encoded vaccine antigen and the epidemic influenza virus, and greatly improve the protective effects of the influenza vaccine. Although there are several disadvantages in the DNA vaccine platform, conserved immunogens developed on the DNA vaccine platform are promising candidates for the creation of the universal vaccine against influenza viruses. Their ability to induce protective immunity has been demonstrated in animal models. Universal influenza virus DNA vaccine candidates are most often designed by adopting genes encoding the conserved proteins HA2, NP, M1, and M2, and the catalytic subunit of PB1 [46,47,48]. The encoding genes in DNA plasmids expressed individual protein antigens or combined protein antigens that could induce cross-protective immunity against homologous or heterologous influenza viruses in animal models [49]. Study results show the efficacy of universal influenza DNA vaccines encoding multiple antigens, conferring better protection than that encoding a single antigen.

Researchers created an optimized M2e DNA vaccine and tested its efficacy against both homologous and heterologous influenza viruses. It induced significant humoral and cellular immune responses, suggesting that this vaccine candidate provided potent protection against both homologous and heterologous viruses [50]. The article by Jaroslav Hollý et al. described studies on M2e DNA vaccines and the results of the immunogenicity and protective properties of the vaccine were evaluated by administering the M2e DNA vaccine to BALB/c mice and performing viral challenge experiments, showing that the mice produced anti-M2e antibodies after vaccination with the M2e DNA vaccine. The survival rate of mice vaccinated with the M2e DNA vaccine was high after the viral challenge experiment [51]. Therefore, such studies indicated that the optimized M2e DNA vaccine could be a promising candidate for a universal influenza vaccine, inducing both humoral and cellular immune responses.

### 3.3. Development of Universal Influenza mRNA Vaccines

As the mRNA vaccine platform has shown great promise in the development of vaccines against infectious diseases, it has also been used to develop universal mRNA vaccines against influenza. In the last decade, research on a universal influenza mRNA vaccine progressed dramatically. The successful employment of SARS-CoV-2 mRNA vaccines in control of COVID-19 further accelerated the research and development of universal influenza mRNA vaccines. Recent universal mRNA influenza vaccines are summarized in Table 1. There are two methods being used for the development of universal influenza mRNA vaccines, using combined antigens or conserved antigens. Combined nucleoside-modified mRNA influenza vaccines including antigens from different subtypes have shown promise as universal influenza vaccines. 

Freyn et al. designed a multi-targeting nucleoside-modified mRNA influenza virus vaccine that provided broad protection against influenza viruses in mice [52]. In 2021, research published by Sudha Chivukula et al., focused on the development of multivalent mRNA vaccine candidates for seasonal or pandemic influenza. The research team aimed to develop a novel influenza vaccine using mRNA technology to provide broader protection by inducing immunity against different influenza virus strains. They used mRNA lipid nanoparticle (LNP) technology to deliver mRNA encoding influenza virus hemagglutinin (HA) and neuraminidase (NA) to stimulate the body to mount an immune response against these viral proteins. The immunogenicity and safety of this mRNA vaccine candidate was evaluated through immunization experiments in mice and rhesus monkeys. The goal of this research was to provide new strategies and methods for developing more effective influenza vaccines [7]. McMahon and coworkers suggested that a quadrivalent nucleoside-modified mRNA influenza vaccine contained four influenza A group 2 virus antigens provided protection against group 2 influenza virus [53]. Pardi et al. reported that a pentavalent nucleoside-modified mRNA vaccine conferred broad protection against influenza B viruses [54]. These results support the concept of nucleoside-modified mRNA-LNPs expressing multiple conserved antigens as universal influenza virus vaccines.
vaccines-12-00664-t001_Table 1Table 1Universal influenza mRNA vaccines that have been published.AuthorsPublished YearAntigensAnimal ModelClinical TrialFreyn et al. [52]2020HA stalk, NA, M2, NP,Mice
Arevalo et al. [55]2022HA from 20 subtypesMice
McMahon et al. [53]2022HA stalk, NA, M2, NPMice
Ven et al. [56]2022NP, M1, PB1Ferret
Zhu et al. [57]2022HA, MI, NPMice
Pardi et al. [54]2022B/Yamagata/16/1988-like lineage HA B/Victoria/2/1987-like lineage HA, NA, NP, and M2Mice
Widge et al. [58]2023HA stabilized stem
Phase 1Lee et al. [16]2023HA of (A/H1N1, A/H3N2, B/Victoria, and B/Yamagata)
phase 1/2Xiong et al. [59]2023M2e, HA stalk, NP,Mice


Subsequently, on 24 November 2022, the team of Drew Weissman and Scott E. Henslsy of the University of Pennsylvania published “A multivalent nucleoside-modified mRNA vaccine against all known influenza virus subtypes” in the top journal *Science* [55]. They developed a multivalent mRNA vaccine against all existing known subtypes of influenza virus. Their results showed that mice inoculated with the multivalent mRNA vaccine were able to trigger antibody responses against all 20 HA antigens, and the multivalent mRNA vaccine elicited high levels of cross-reactivity and subtype-specific antibodies in mice and ferrets, and maintained high levels for 4 months, thus protecting the host from infection with matched and mismatched strains of influenza virus. This study fully confirms that mRNA vaccines can provide protection against antigenic variants by simultaneously inducing antibodies against multiple antigens.

In recent research, adjuvant was introduced to universal influenza virus vaccines and enhanced the immune response. Zhu et al. designed cGAMP-adjuvanted multivalent influenza mRNA vaccines that induced broad protective immunity through cutaneous vaccination in mice [57].

In the last year, Li Xiuling and colleagues published their results in *Emerging Microbes & Infections* with the title of “An mRNA-based broad-spectrum vaccine candidate confers cross-protection against heterosubtypic influenza” [59]. In this study, they developed a multivalent influenza mRNA vaccine based on influenza protective antigens, consisting of three conserved antigens of influenza A virus, including the extracellular domain (M2e) of the M2 ion channel, the long α helix (LAH) and nucleoprotein (NP) of the hemagglutinin stem region, with the aim of enhancing its efficacy and facilitating the development of future broad-spectrum influenza vaccines. The immunogenicity of this M2e-LAH-NP influenza mRNA vaccine was evaluated in a mouse model. Their results showed that it was effective in triggering serum antibody responses and cellular immune responses against the three protective antigens, also induced antibody-dependent cell-mediated cytotoxic effects and cross-reactive CD8+ T cell immune responses, and conferred broad protection against the H1N1, H3N2, and H9N2 viruses. In addition, single-cell transcriptional analysis of T cells in the spleen of inoculated mice showed that it significantly promoted the differentiation of CD8+ T cells and memory T cells through prime-boost immunization. This study illustrated that mRNA influenza vaccines encoding conserved proteins are a very promising strategy to trigger broad protective humoral and cellular immunity against various influenza viruses.

## 4. Conclusions

Since the concept of universal influenza vaccines was coined, research has advanced dramatically in this field. The development of a universal influenza vaccine that provides broad and durable protection against multiple epidemic and emerging influenza viruses is a long-term goal of public health and pandemic preparedness. Universal influenza DNA vaccines have been the subject of many studies over the past decades due to their ability to induce humoral and cellular immune responses in various animal models. Although DNA vaccines have shown intrinsic defects, such as integration of their DNA into the host genome and low efficiency in large animals and humans, so far, the data available have alleviated these concerns. Genome insertion could be controlled with modified DNA sequences and their low immunogenicity in humans could be improved with several strategies [41]. Given these advantages, influenza DNA vaccines have the potential to provide a broader protection as universal vaccines with the characteristics of easy preparation, simplicity of formulation, high stability and safety by lacking infectious regents [60].

The mRNA vaccine platform holds promise to improve influenza vaccination by improving strain matches, enclosing multivalent antigens or conserved antigens, and inducing broader humoral and potent cellular immune responses. However, work on a universal influenza vaccine based on mRNA technology has just begun. Furthermore, as a new biotechnology, many regulations will be demanded and its accompanied uncertain safety and vaccine hesitancy need to be resolved [61,62]. However, promoted by the success of the SARS-CoV-2 mRNA vaccines, the advent of mRNA technology has made it possible to see the dawn of a universal influenza vaccine, despite that there are still many technical difficulties to be overcome, whether it is a multivalent mRNA vaccine using the exhaustive existing HA antigens or a multivalent mRNA vaccine based on the same conserved antigen combined with other conserved antigens; for example, the formulation of LNP formulations for encapsulated multivalent mRNA vaccines or the structure optimization of multivalent antigen fusion proteins, the development of more efficient delivery vectors, and the addition of novel adjuvants [63]. So far, the available data suggest that mRNA-based universal influenza vaccines could be improved and hold great promise. Thus, it is optimistic that under the endeavors of experts from academic institutions and industry entities, the successful breakthrough of a universal influenza vaccine will not only open up a new situation in the research and development of viral vaccines, but also consolidate the cornerstone role of mRNA technology in the field of vaccine development in the near future.

## Figures and Tables

**Figure 1 vaccines-12-00664-f001:**
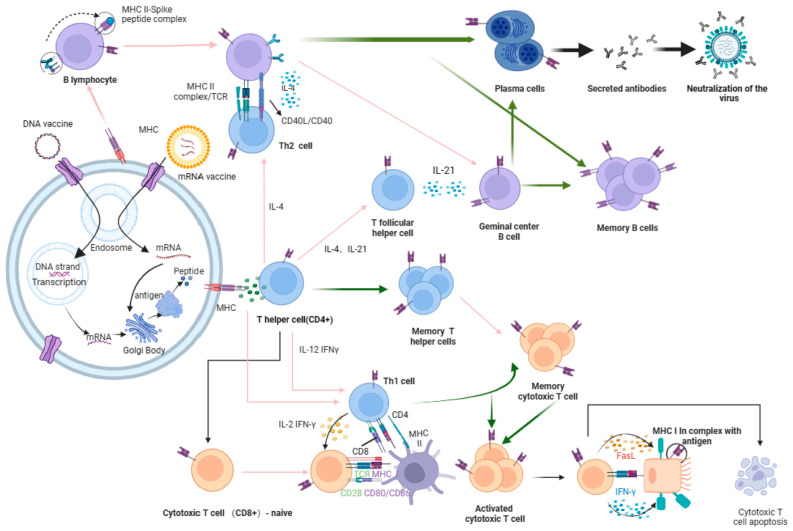
The immune response mechanisms of different nucleic acid vaccines, including DNA vaccines and mRNA vaccines. Briefly, after encapsulated DNA or mRNA is released from endosomes into the cytoplasm, the protein encoding T- or B-cell epitopes is produced and digested to generate antigenic peptides to be presented by the major histocompatibility complex (MHC) on the cell surface, recognized by receptors on the ancestral B cells or T cells, eliciting humoral and cellular immune responses. Humoral immunity is activated through the synergistic action of T helper (Th) cells via the MHC II pathway. Activated T cells secrete cytokines, promoting B cells to differentiate into plasma cells or memory B cells in the germinal center. They secrete specific antibodies to neutralize influenza viruses. Helper T cells enhance the overall immune response by recognizing antigens and producing cytokines via the MHC II pathway, while CD8+ cytotoxic T cells recognize and destroy virus-infected cells expressing pathogen-specific antigens via the MHC I pathway. MHC, major histocompatibility complex.

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
