# Peer review of "Advances in Nucleic Acid Universal Influenza Vaccines"

_vaccines, 2024, doi:10.3390/vaccines12060664_

Round 1

Reviewer 1 Report

Comments and Suggestions for Authors

Xu et al has summarized the current advances in nucleic acid universal influenza vaccines. The review provides a succinct summary of current studies of nucleic acid and universal influenza vaccines. However, the review lacks some discussion point that will be vital for future deployment of nucleic acid influenza vaccine and “universal” vaccines.  I have the following comments:

Major comments:

1.      The section on influenza typology and structural characteristics is too long. For a short review aiming to focus on nucleic acid, almost half of the review was spent talking about influenza genomic structure and protein functions. This section should be summarized and more details on vaccine and nucleic acid vaccine portion should be given.

2.      There should be a section first detailing the current influenza vaccines, their shortcomings, and how nucleic acid vaccines can help resolve these issues. This section should come before detailing the current developments in DNA and mRNA universal influenza vaccines.

3.      There should be section discussing the potential of deploying nucleic acid vaccine as mucosal vaccines against influenza virus, collating studies pertaining to this topic, and challenges associated with mucosal nucleic acid vaccine delivery, as there is a high interest in mucosal vaccines for respiratory viruses.

4.      There are currently phase 1/2 clinical trials (e.g. PMID: 37336877) already ongoing for mRNA based seasonal influenza vaccines. While these are not universal influenza vaccines, with the authors stating that nucleic acid vaccines are fast to produce in large quantities, one would argue against the need of a universal influenza vaccine, that will be hard to fully validate their “universal” cross protection against circulating strains, unless followed longitudinally over long periods of time, in large scale clinical trials, which will be highly resource intensive and hard to justify the economy of scale. As such, the authors should discuss how universal nucleic acid vaccines is still desirable as compared to producing seasonal mRNA vaccines, which will be updated to ensure maximal protection annually.  

5.      The authors should have a table summarizing all the studies shown, their current status, their efficacy, current stage of trials etc.

Minor comments:

1.      The manuscript needs language proof-reading. There are a number of grammar, sentence structure and spelling errors in the manuscript.

2.      Figure 1 legend is listed as “figure legend” instead of as “figure 1”.

Comments on the Quality of English Language

 The manuscript needs language proof-reading. There are a number of grammar, sentence structure and spelling errors in the manuscript.

Author Response

Dear reviewer,

Thank you so much for you comments and suggestion. We have revised the manuscript thoroughly. Any comments will be appreciated.

Reviewer 2 Report

Comments and Suggestions for Authors

This is an important topic to review as universal vaccine for influenza is as the authors state a “long-term goal of public health and pandemic preparedness”. The authors have described the influenza virus, nucleic acid vaccines, universal influenza vaccine and universal influenza mRNA vaccines. However, the review is very limited and looks more like a summary of few papers from the literature and not a focused review which provides a robust discussion of the subject matter. The manuscript should be revised to enhance its depth and scope before it can be considered for publication. Below are my suggestions and comments for revision:

1.     Abstract: broad-spctrum influenza-typo in the abstract itself shows that the manuscript was not proofread prior to submission.

2.     “Compared to conventional influenza vaccines that composed protein or live-attenuated influenza viruses, nucleic acid influenza vaccines inducing potent humoral and cellular immune response enhance the protection against a broad array of influenza viruses(Melo et al., 2022).” -this statement is confusing, do the authors mean that LAIV do not produce humoral and cellular immune response? LAIV does indeed show robust humoral and cellular immune response. Moreover, the paper that is cited, does not fully support this statement.

3.     The authors give a lot of details on Influenza virus typology and structural characteristics, this can be made a bit more concise as this is not the key topic of this review. More emphasis could be given to the development of nucleic acid vaccines, discussing more literature references beyond the one cited.

4.     The advantages and disadvantages of nucleic acid vaccines should also be elaborated upon, with additional references from the literature.

5.     The section titled "Development of Universal Influenza mRNA Vaccines" summarizes several papers but could benefit from a more focused discussion on the development of a universal influenza vaccine. It would be useful to elaborate on why the development of a universal vaccine using mRNA or other nucleic acids is desirable.

6.     The manuscript has a lot of deficiencies in grammar, particularly tense usage, and English language utilization. It is recommended that the manuscript be extensively revised to meet the required standards.

Comments on the Quality of English Language

The manuscript has a lot of deficiencies in grammar, particularly tense usage, and English language utilization. It is recommended that the manuscript be extensively revised to meet the required standards.

Author Response

(The authors gave the same response as above.)

Reviewer 3 Report

Comments and Suggestions for Authors

The authors have submitted the review manuscript entitled “Advances in Nucleic Acid Universal Influenza Vaccines”. In the manuscript, the authors discuss the recent advances in DNA and RNA-based universal influenza vaccines.

Even though the topic is worth summarizing, the manuscript is very poorly written with respect to the quality of the English language, description and inclusion of relevant research, and relevant references. The authors have not analyzed the potential gaps in the research and have not summarized nor mentioned the available literature. I have several concerns as given below:

1.       In the very first line of the manuscript the authors give the statistics given by WHO but do not cite WHO.

2.       The authors then refer to the influenza pandemics but only mention the 1918 and 2009 pandemics, even though there have been 5 reported influenza pandemics till date.

3.       The authors then talk about the strategies employed to eliminate the defects of existing influenza vaccines, however, there is no mention at all of what defects are they referring to and what are the implications of those defects.

4.       The authors then talk about the development of vaccines covering as many strains as possible. However, each year the quadrivalent influenza vaccine is made to match the currently circulating influenza virus strains (H1, H3, B Victoria and B Yamagata). It is unclear whether the authors are referring to the occasional mismatch between vaccine strains and circulating strains or the inclusion of additional historical strains in the vaccine and what will be the advantages of doing so.

5.       The authors give a figure that illustrates the immune response induced by nucleic acid-based influenza vaccines. However, there is not description of this throughout the manuscript. Moreover, the authors mention that nucleic acid-based influenza vaccines induce more potent humoral and cellular immune response as compared to protein or inactivated virus based influenza vaccines. However, the difference in the induction of the immune response and the potential reasons are not given.

6.       There is not even a single reference included in the influenza epidemiology section.

7.       The authors do not mention people with co-morbidities in the high-risk group.

8.       The authors talk about anti-influenza drugs, which seems out of place and irrelevant in the topic.

9.       The authors describe (with no relevant references at all) the influenza A virus genome and the fact that IBV genome varies from IAV, they do not describe the key features of IBV genome.

10.   The authors discuss the role of M1 and M2 influenza viral proteins, but there is no mention or discussion of any of the other influenza proteins.

11.   In the nucleic acids vaccines section, the authors give a very irrelevant introduction of these vaccines, no specific information is mentioned for influenza vaccines.

12.   Similarly the advantages and disadvantages section is very vaguely written. The manuscript needs a thorough and in-depth discussion of the advantages and drawbacks of these vaccines with respect to influenza.

13.   Finally in the universal influenza DNA vaccines and mRNA vaccines sections, the authors only list the results of 3 studies (in each section). It looks more like an abstract of each study given one after the other rather than a review of the relevant literature.

Comments on the Quality of English Language

Extensive English editing is needed. I suggest getting the manuscript corrected by a native English speaker.

Author Response

(The authors gave the same response as above.)

Round 2

Reviewer 1 Report

Comments and Suggestions for Authors

The authors have addressed most of my comments. However, while the contents of the manuscript are improved, the manuscript still requires major formatting and language editing. Much of the sentence structure and grammar inaccuracies makes the manuscript hard to follow, which will be a pity considering there is important information captured in this manuscript. I have the following comments.  

1.       The write up in line 128 to 130 “There are also reports on the spillover of avian influenza viruses from avian to human, such as constantly reported infection by H51N1 influenza viruses with the potential human-to-human transmission, H7N9 epidemic in 2013 caused over 1500 infections, H3N8 infections in human etc.”, please provide the references to back these statements.

2.       In section 1, the introduction text should only be up until line 48. Anything after that (line 49 to 115), and figure 1 belongs in section 3. The write-up after line 48 should be shifted and integrated into section 3 (at lines 186 to 202, which this subsection feels like a summarize version of this part of the writing in section 1), as a subsection to introduce nucleic acid vaccine and universal nucleic acid vaccine, to ensure the flow of the writing.

3.       Section 2.2 is still too long. It is suggested that the section should further condensed the known influenza biology portion, and focus on those proteins that are vital antigens for vaccine production (e.g. the new text addition from line 170 to 183 should be the focus of this section/subsection)

4.       In my previous comments (point 4), I requested the reviewers to comment and justify on the need to have universal nucleic acid vaccine and provide their advantages over rapid production of seasonal nucleic acid vaccines. The authors only mentioned in passing that there are clinical trials testing on nucleic acid vaccines, but did not provide elaboration to justify advantage of deploying universal vaccine, compared to the current seasonal vaccine update practices. There are some loose writeup at different parts of the manuscript that mention cross protection into avian viruses, which I think is an important point, among others. I would like to see this be elaborated more in the manuscript (e.g. as an additional, short subsection in section 3 discussing this, before the conclusion section).

5.       The language still requires major editing to aid with the flow of the manuscript. Current sentence structure can be challenging to decipher in many parts of the manuscript. Some examples of incorrect term usage, and grammatically or structurally incorrect sentences are as below:

“…influenza vaccines is still effective strategy to prevent seasonal infection…”

“…including seasonal influenza viruses and emerging zoonosis influenza viruses”

“Nucleic acid vaccine were divided into DNA vaccines and RNA vaccines”

“… the crucial strategy to prevent pandemic or endemic of influenza viruses”

“Seriously, inactivated and recombinant influenza vaccines fails to trigger a vigorous immune response…”

“Egg-based influenza vacciens often caused antigen change”

“there is a time gap between the influenza viruses recommended by WHO influenza surveillance system and vaccines which caused the prompt influenza vaccine supply after the influenza pandemic or epidemic brought out…”

“As shown, after encapsulated DNA or mRNA entering target cells…”

“…essential for viral repli-150 cation and protect viral RNA frombe degradation by RNases”

There are many more similar language inaccuracies in the whole manuscript. The revised manuscript should not contain these writing inaccuracies, and I implore the authors to take this seriously. It is suggested to have the entire manuscript proofread for language by native English language speakers to improve the readability of the manuscript.

Comments on the Quality of English Language

See point 5 of comments and suggestion above

Author Response

Dear reviewer,

Thank you so much for your review. We have revised this manuscript according you suggestions.

Best Wishes

Reviewer 2 Report

Comments and Suggestions for Authors

The response to the comments are adequate and the revised manuscript is saftisfactory.

Author Response

Thank you so much for review!

Reviewer 3 Report

Comments and Suggestions for Authors

The authors have submitted a revised review article titled "Advances in Nucleic Acid Universal Influenza Vaccines".

Even though the manuscript is better than the first version, it still has several flaws which need to be addressed.

1. The manuscript does not cite appropriate references. Given the vast majority of literature available on influenza vaccines, it is very unbelievable that the authors have cited only 30 papers in the review article. A major part of the manuscript (about 75-80%) cites some information without any reference to it.

2. My previous comment (comment no. 4) has not been addressed even though the authors mention that the section has been revised.

3. In lines 45-46, the authors mention that the egg-based vaccines cause 'antigen change'. It is unclear what the authors mean here by the term 'antigen change'. Are they referring to viruses evolving in the eggs? or the antigenic shift and drift?

4. In the same line, the authors talk about the 'time gap', which is again very vague and very difficult to interpret what authors are talking about.

5. In lines 55-59, the authors compare the conventional vaccines with the nucleic acid vaccines based on their antigens. Here they write the antigen in the conventional vaccine but do not mention the antigen in nucleic acid influenza vaccines.

6. The figure legend to the figure 1 should be concise. Here what authors write as figure legend is simply the repetition from the earlier text.

7. Again, in the figure legend the authors write that the DNA is translated into antigenic protein, but do not mention what is the antigenic protein.

8. In line 190, the authors mention that the DNA vaccines encode 'interesting antigens'. This is a very vague and flawed term that should not be used in a scientific paper.

9. In lines 225-226, the authors mention that 'once the standard procedure has been set, the manufacture process of DNA vaccines is very simple'. Is the standard procedure been established? Is it for the influenza vaccines? If for some other vaccines, what are the implications specific to influenza? Again, no reference is given anywhere!

10. The universal influenza DNA and RNA vaccines should be described in more detail as it is the main focus of the review.

11. The English language is still flawed and there are several typos throughout the manuscript. I have given some of these below:

a. Line 16: showed great promising in

b. Line 66: transported into nucleus firstly

c. Line 62: particles is

d. Line 82: cytokines has

e. Line 127: symptoms is

f. Line 204: highly safety

Comments on the Quality of English Language

The English language needs major editing, as described above.

Author Response

Dear Reviewer,

Thank you so much for your review. We have revised this manuscript according your suggestions.

Best Wishes

Round 3

Reviewer 1 Report

Comments and Suggestions for Authors

The authors have taken my comments into consideration and revised the manuscript sufficiently. There are still some minor typographical errors, which I believe can be fixed during typesetting by the journal editors.

Comments on the Quality of English Language

There are still some minor typographical errors, which I believe can be fixed during typesetting by the journal editors.

Author Response

Thank you so much for your support on our work!

Reviewer 3 Report

Comments and Suggestions for Authors

The authors have submitted a further revised version of the review article titled "Advances in Nucleic Acid Universal Influenza Vaccines". The manuscript still needs further work before it can be considered for publication.

I have described my concerns below:

1. Even though the authors have cited a few more references as compared to the previous version, the manuscript still has a lot of information without any reference given to it, examples described below:

a. The entire section of the mechanism of nucleic acid vaccines (lines 62-116) has only 1 reference to it. It is absolutely unbelievable that there is only 1 published paper that describes the mechanism of DNA and mRNA influenza vaccines.

b. There is not a single reference to the disadvantages of nucleic acid vaccines (lines 211-217)

c. In the universal influenza DNA vaccines section: (i) lines 220-235 cite a lot of information without any reference (ii) lines 236-247 again have a lot of information with only 1 reference.

d. The entire Table 1 has names of authors and published year but no reference of the published paper!

e. The entire conclusion section (lines 313-344) again has no reference given!

f. In line 190-191, the authors talk about the adjuvants used in DNA vaccines (without reference to any of the given adjuvants).

g. In the same section, the authors mention the routes of administration of DNA vaccines, again without any references!

2. The manuscript still has a flawed English language in several places. Examples described:

a. Line 50: an report

b. Line 57: adjuvants showed cross-reactive in broad manner

c. Line 129: recover with 1-2 weeks

d. Line 191: CpG et al.

e. Line 212: DNA vaccine is relatively low immunogenicity

f. Line 255: used for develop

g. Line 257: showed promising to be

3. As pointed out in my previous comments, the authors still talk about 'interesting antigens' (line 95) but do not describe what antigens (which viral proteins/ epitopes) they are referring to!

4. In lines 188-189, the authors again talk about the plasmids that encode antigens that stimulate humoral immune response. What are these antigens? Are these HA proteins? NA proteins? NP proteins? or some other viral proteins? What has been reported in the published papers?

5. Even though the authors mention that they have deleted the error (as pointed out in my previous comments) they still write in line 204 "The construction of nucleic acid vaccines is simple once the standard procedure is established". If there is no standard procedure, how do they know if it is simple or not? Do not make such haywire statements without citing any reference!

Comments on the Quality of English Language

The English language still needs editing as described in my comments.

Author Response

Dear Reviewer,

Thank you so much for support on our manuscript. We appreciate you so much!

Round 4

Reviewer 3 Report

Comments and Suggestions for Authors

The authors have addressed most of my concerns regarding the manuscript. However, I had mentioned that there are several places which had flawed English language. I only pointed out a few of them as an example. Looks like the authors made corrections only where I pointed out the flawed English language. The entire manuscript needs to be thoroughly checked for quality of English language and the typos.

Comments on the Quality of English Language

The manuscript needs English language and typos corrections. I suggest getting this checked by some native English language speaker.

Author Response

Dear Reviewer,

Thank you so much for your patience and suggestion on  our manuscript. We have proofed this manuscript carefully. If you have any suggestion, please let me know.

Best Wishes

Junwei